# Gut microbiota induces high platelet response in patients with ST segment elevation myocardial infarction after ticagrelor treatment

Xi Zhang[1,2†], Xiaolin Zhang[1†], Fangnian Tong[1], Yi Cai[1], Yujie Zhang[1], Haixu Song[1], Xiaoxiang Tian[1], Chenghui Yan[1], Yaling Han[1,2]*

[1]Department of Cardiology and Cardiovascular Research Institute of PLA, General Hospital of Northern Theater Command, Shenyang, China; [2]Department of Cardiology, Shengjing Hospital of China Medical University, Shenyang, China

*For correspondence:
hanyaling@163.net

[†]These authors contributed equally to this work

## Abstract

**Background:** Ticagrelor is a first-line drug for the treatment of acute ST elevation myocardial infarction (STEMI). However, approximately 20% STEMI patients taking ticagrelor exhibited a delayed response and the mechanism was still unclear.

**Methods:** To explore the mechanism of the poor response of ticagrelor in post-percutaneous coronary intervention (PCI) patients, we enrolled 65 high platelet reactivity (HPR) patients and 90 controls (normal platelet reactivity [NPR]). Pharmacokinetic assessment result showed that the plasma concentrations of ticagrelor and its metabolism production, AR-C124910XX, were lower in HPR patients than controls. Further single nucleotide polymorphism (SNP) analysis identified that there is no difference in ATP binding cassette subfamily B member 1 (*ABCB1*) gene expression between the NPR group and the HPR group. Metagenomic and metabolomic analyses of fecal samples showed that HPR patients had higher microbial richness and diversity. Transplantation of the gut microbiota from HPR donors to microbiota-depleted mice obviously decreased plasma concentration of ticagrelor.

**Results:** Our findings highlight that gut microbiota dysbiosis may be an important mechanism for the ticagrelor of HPR in patients with STEMI and support that modify gut microbiota is a potential therapeutic option for STEMI.

**Conclusions:** Our findings highlight that gut microbiota dysbiosis may be an important mechanism for the ticagrelor of HPR in patients with ST elevation myocardial infarction (STEMI) and support that modify gut microbiota is a potential therapeutic option for STEMI

**Funding:** NSFC 82170297 and 82070300 from the National Natural Science Foundation of China.

## Editor's evaluation

The authors investigated the possible mechanisms of inadequate platelet inhibition with ticagrelor. The high platelet reactivity was found to be associated with lower plasma concentrations of ticagrelor and its active metabolite as compared to the control group. The idea of a two-step research provided key support for clinical findings with experimental data suggesting that gut microbiota dysbiosis may be an important mechanism for the high platelet reactivity in patients treated with ticagrelor. This scientific approach rises new ideas for further research on potential modifications of the gut microbiota as a new therapeutic option in patients with insufficient platelet response to ticagrelor.

## Introduction

Presently, the standard of treatment in ST elevation myocardial infarction (STEMI) patients includes dual antiplatelet therapy consisting of aspirin plus a P2Y12 receptor antagonist. The currently used P2Y12 receptor inhibitors are clopidogrel, prasugrel, and ticagrelor. Moreover, clopidogrel and prasugrel require metabolic activation and irreversibly bind to the P2Y12 receptor, causing prolonged recovery of platelet function. In addition, the efficacy of clopidogrel is reduced in patients with certain genotypes. Ticagrelor is a new oral antiplatelet agent of the cyclopentyltriazolopyrimidine class which acts through the P2Y12 receptor. In contrast to clopidogrel and prasugrel, ticagrelor does not require metabolic activation and binds rapidly and reversibly to the P2Y12 receptor.

However, it undergoes cytochrome P450 (CYP)-mediated metabolism to produce the active metabolite AR-C124910XX (*Yandrapalli et al., 2019*). AR-C124910XX, the main known active metabolite, has approximately 30% activation of the parent drug to exhibit antiplatelet activity (*Mehta et al., 2010*). As it quickly and strongly came into effect compared to the conventional drug clopidogrel, the latest American College of Cardiology and the American Heart Association guidelines recommend ticagrelor as a first-line drug for the treatment of acute STEMI (*Levine et al., 2016*).

However, some STEMI patients respond poorly to ticagrelor, exhibiting a delayed response in the early phase of treatment even when a loading dose of 180 mg was employed. There is a crucial time window after percutaneous coronary intervention (PCI) that cannot be protected by ticagrelor, in which some STEMI patients are at a high risk of stent thrombosis (*Alexopoulos et al., 2012*; *Bagai et al., 2018*; *Franchi et al., 2015*). The mechanism of high platelet reactivity (HPR) in ticagrelor remains elusive. Genetic and non-genetic factors have been reported to contribute to its HPR (*Siller-Matula et al., 2016*; *Barbieri et al., 2016*; *Verdoia et al., 2016*).

Ticagrelor requires absorption in the intestine combined with the P2Y12 receptor of platelets without metabolic activation in the liver (*Weeks et al., 2015*). Transporters are transport proteins that mediate transmembrane drug transport in the intestine. P-glycoprotein is encoded by ATP binding cassette subfamily B member 1 (*ABCB1*) gene. As an important transporter, it affects the absorption and metabolism of ticagrelor in the intestine (*Varenhorst et al., 2015*; *Notarangelo et al., 2018*). P-glycoprotein, also known as ABCB1 protein, is one of the adenosine triphosphate binding cassette genes that mostly exist in the apical membrane of the intestinal mucosa, liver, and kidney (*von Richter et al., 2004*; *Fromm and Kim, 2011*). It encodes transporters and channel proteins, acts as an efflux pump, and is responsible for intracellular homeostasis. *Marsousi et al., 2016*, revealed a significant inhibitory effect of the P-glycoprotein inhibitor valspodar on the efflux of ticagrelor, suggesting that P-glycoprotein is involved in the oral disposal of ticagrelor. A large genetic substudy of the PLATO trial showed that *CYP2C19* and *ABCB1* polymorphisms were independent of the lower rates of cardiovascular death, MI, or stroke observed in patients treated with ticagrelor (*Varenhorst et al., 2015*; *Nardin et al., 2018*; *Wallentin et al., 2010*). In addition, a genome-wide association study of patients from the PLATO study identified a single nucleotide polymorphism (SNP) associated with the solute carrier organic anion transporter family member 1B1 (*SLCO1B1*) gene that leads to decreased organic anion transporting polypeptide 1B1 (OATP1B1) activity. Plasma ticagrelor levels were associated with two independent SNPs in the *CYP3A4* region and an SNP in the UDP-glucuronosyltransferase 2B7 (*UGT2B7*) gene (*Varenhorst et al., 2015*). However, the detailed mechanism remains unclear.

There is growing awareness that alterations of the gut microbiome and its products may affect the progress and state of cardiovascular disease (*Zhernakova et al., 2018*; *Jie et al., 2017*). The association between gut microbiota and ticagrelor has not been reported previously, and the specific components of microbiota that influence the effect of drugs leading to the occurrence of HPR remain unknown. In this study, we aimed to investigate the association between gut microbiota dysbiosis and drug absorption and a certain genus of gut microbiota that might be the cause of delayed absorption of ticagrelor.

## Materials and methods
### Study cohort and patient characteristics

All patients with STEMI were enrolled in the Department of Cardiology of the General Hospital of Shenyang Military Region between April 2016 and March 2018. All patients (the onset time for STEMI is ≤12 hr) received ticagrelor (180 mg loading dose followed by two 90 mg maintenance doses) and

aspirin (300 mg loading dose followed by a 100 mg maintenance dose) before PCI. Briefly, STEMI patients were clinically diagnosed using a combination of the following criteria: (1) acute ischemic-type chest pain in the last 24 hr; (2) electrocardiogram change (pathological Q wave, ST segment elevation, or depression); (3) elevated plasma hscTnT (0.05 ng/mL) (*Ibanez et al., 2018*). The exclusion criteria were as follows: allergy/intolerance to aspirin or ticagrelor; use of oral anticoagulants or antiplatelet agents other than aspirin and ticagrelor; recent treatment with a glycoprotein IIb/IIIa antagonist; use of proton pump inhibitors; end-stage renal or hepatic disease; treatment with fibrin-specific fibrinolytic therapy <24 hr or non-fibrin-specific fibrinolytic therapy <48 hr prior to randomization; presence of active internal bleeding or history of ischemic or hemorrhagic stroke in 6 months; platelet count <100 × $10^9$ /L; hematocrit <25%; creatinine levels < 2.5 mg/dL; hepatic disease (hepatic enzymes twice the upper normal limit). The use of human sample complies with the Declaration of Helsinki and was approved by the ethics committee of the General Hospital of Shenyang Military Region, and written informed consent was obtained from each subject. A flow diagram of this study is presented in *Figure 1—figure supplement 1*.

## Detection of platelet aggregation rate

According to the standard protocol, the rate of platelet aggregation (PA) was determined using light transmission aggregometry. Peripheral venous blood samples were collected with a loose tourniquet through a short catheter inserted into the forearm vein. The first 2–4 mL of blood was discarded to avoid spontaneous platelet activation, and the rest was collected in 3.2% trisodium citrate for PA measurement. The upper platelet-rich plasma was obtained by centrifugation at 1000 rpm for 10 min at room temperature. The remaining blood was centrifuged at 3000 rpm for 10 min to obtain platelet-poor plasma (PPP). Twenty-five microliters of ADP (20 µM) was used as a platelet inducer. Following addition of PPP into the platelet reaction cup, the PA rate was measured at a speed of 1000 rpm. The PA rate was measured at 0, 2, 4, 6, 8, 12, and 24 hr after administration of the 180 mg ticagrelor. According to the rate of PA at 2 hr, patients were divided into two groups: HPR group (PA ≥59%) and normal platelet reactivity (NPR) group (PA <59%).

## Pharmacokinetic assessment

Blood samples for the determination of plasma concentrations of ticagrelor and AR-C124910XX were collected at 0 hr (pre-dose) and at 2, 4, 6, 8, 12, and 24 hr after ticagrelor administration. Blood was collected from the antecubital vein into Vacutainer tubes containing 2 mg/mL K2-EDTA. The plasma concentrations of ticagrelor and AR-C124910XX were determined using the HPLC-MS/MS system as previously described by *Sillén et al., 2011*; *Sillén et al., 2010*. Samples were analyzed within 3 months of collection, well within the known period of stability. The maximal concentration (Cmax) and time for the maximal plasma concentration to reach Cmax (Tmax) were calculated directly from the measured plasma concentration of each patient and presented as medians with interquartile ranges.

## Genotyping

In the past few decades, scholars have found that *ABCB1* genetic variation has a great impact on *ABCB1* gene expression and function, especially in SNPs. Among them, rs2032582 (*G2677T/A*), rs1045642 (*C3435T*), and rs1128503 (*C1236T*) have attracted wide attention because of their highly variable frequency and strong linkage disequilibrium in different populations (*Martinelli et al., 2014*; *Pontual et al., 2017*; *Raymond et al., 2021*). The SNP gene sequences were searched for and downloaded from the NCBI public database (http://www.ncbi.nlm.nih.gov). Primer 5.0 software was used to design polymerase chain reaction (PCR) primers (see *Table 1—source data 1* for details). Primers were synthesized by Shanghai Bioengineering Technology Co., Ltd. Genomic DNA was extracted from 3 mL of peripheral blood leukocytes using the standard phenol/chloroform method. The genotypes and allele frequencies of *ABCB1* SNPs were detected using PCR and pyrosequencing.

## Stool sample collection and DNA extraction

Each study participant was asked to collect fecal samples in the morning using fecal collection containers. Fecal samples were stored at 4°C immediately after defecation, brought to the laboratory within 4 hr, and stored at –80°C before DNA extraction. The containers were transferred on ice and stored at –80°C prior to processing. Total bacterial DNA was extracted from fecal samples using the

QIAamp DNA Stool Mini Kit (QIAGEN, GmbH, Germany) according to the manufacturer's instructions. DNA purity was measured using a Thermo NanoDrop 2000 Spectrophotometer (Thermo Fisher Scientific, Waltham, MA). All DNA extraction procedures were prepared under a Class II biological safety cabinet. The concentration of genomic DNA in each fecal sample was quantified using the NanoDrop 2000 Spectrophotometer. DNA integrity and size were assessed using 1% agarose gel electrophoresis. For quality control, restriction digestion was performed to screen these polymorphisms in 100 randomly selected samples, which showed 100% concordance.

## 16S RNA gene amplicon and sequencing

Universal primers (341F and 806R) linked with indices and sequencing adaptors were used to amplify the V3-V4 regions of the 16S rRNA gene. Following the preparation of gut microbiota DNA, the library was constructed on the HiSeq 2500 PE250 amplicon sequencing platform, and paired-end sequencing was performed on the Illumina platform. Finally, bioinformatics analysis was carried out using Pandaseq software and Usearch software.

## Metabolomic profiling/$^1$H NMR metabolomics analysis

Three-hundred microliters blood sample and 600 μL methanol were added into EP tube and placed at −20°C for 20 min. After centrifugation at 11,000 rpm for 30 min to take supernatant, methanol was removed with rotary evaporator (SC110A, Thermo, Germany). Then the powder sample was obtained with freeze-drying apparatus and redissolved in 0.55 mL Na$^+$/K$^+$ buffer (0.1 M, 50% D$_2$O, 0.001% TSP). After vortex shaking for 30 s, the sample was well mixed. After centrifugation (4°C, 12,000 rpm, 10 min), 0.5 mL of supernatant was taken and transferred to 5 mm nuclear magnetic tube for inspection. The 1D $^1$H NMR spectra of all samples were collected on a Bruker AVIII 600 MHz NMR spectrometer (Bruker Biospin, Germany) equipped with an ultra-low temperature probe. The proton resonance frequency was 600.13 MHz and the experimental temperature was 298 K.

## Fecal microbiota transplantation

Eight-week-old C57BL mice were purchased from Southern Animal Model Co., Ltd. (Nanjing, China) and placed in ventilated cages separately under a 12 hr light-dark cycle. Mice were fed with SPF-level irradiated mouse chow (Mao Hua Biology, China) and autoclaved water at a constant temperature of 21–22 °C, and maintained at a humidity of 55% ± 5%. Mice were first provided a normal diet for 1 week to adapt to the environment before being randomly assigned to two groups.

Mixed antibiotic cocktails (0.5 g/L vancomycin, 1 g/L neomycin sulfate, 1 g/L metronidazole, 1 g/L ampicillin) that were previously proven sufficient to deplete all detectable symbionts were added to drinking water for 2 weeks to construct a microbiota-depleted mouse model. Thereafter, all mice were freely fed sterile food and water, and bacterial contamination was monitored via regular fecal examination.

For microbiome transplantation, fresh fecal samples were collected from donors, resuspended in sterile saline, and the supernatant was centrifuged. Microbiota-depleted mice were randomly divided into two groups. The fecal supernatant (200 μL) was orally administered to patients in the NPR or HPR group twice weekly. Recipient mice carrying microorganisms were placed in different cages. Drinking water and animal feed were strictly sterilized and replaced daily. The squirrel cage and padding were sterilized with high-pressure steam before use. Padding was changed every 2 days to strictly control bacterial pollution. Blood concentrations of ticagrelor and AR-C124910XX were measured 8 weeks after transplantation.

## Statistical analysis

The statistical analysis of data was carried out using SPSS 22.0. Measurement data were expressed as the mean ± standard deviation. Comparison of the mean number of samples in NPR and HPR was conducted using an independent sample t-test. Count data were expressed as 'number of cases (percentage)'. Comparisons between the two groups were performed using the $\chi^2$ test or Fisher's exact test. The trend of PA at different time points between the two groups was compared using repeated measurement ANOVA. Biological analysis of gut microbiota primarily relies on QIIME, Usearch, Pandaseq, krona, speccum package, and the corrplot package of R software for statistical analysis of sequencing data.

# Results

## Baseline characteristics and platelet reactivity

To compare the difference in PA at different time points before and after taking ticagrelor, we included 155 patients with STEMI, tested patients' PA before (0 hr) and after taking ticagrelor (2, 4, 6, 8, 12, 24 hr). According to the rate of PA at 2 hr, patients were divided into two groups: HPR group (PA ≥59%) and NPR group (PA <59%). There was no significant difference in sex composition, age, weight, and past medical history between the two groups (p > 0.05, *Figure 1—source data 1*). Before taking ticagrelor (0 hr), there was no significant difference between NPR group and HPR group (p = 0.118). The PA of NPR group and HPR group were 66.0% ± 23.5% and 73.4% ± 20.4%, respectively. After ticagrelor administrated, the PA at each time point was lower than that at 0 hr (p < 0.05). And the PA of patients in the HPR group was higher than that in NPR group at 2, 4, 6, and 8 hr (the p-value at each time point <0.001) (*Figure 1A and B*). The decreasing range of PA in the HPR group was lower than that in the NPR group, and did not show the lowest value within 24 hr in the HPR group. In the NPR group, the PA decreased the most at 2 hr after ticagrelor administration, and the lowest value (27.72 ± 16.17) of PA was detected at 4 hr after medication administration.

## Reduction of plasma concentrations of ticagrelor and AR-C124910XX in the HPR group

To compare the difference in absorption efficiency and metabolism of ticagrelor between the HPR group and the NPR group, we measured the concentrations of ticagrelor and AR-C124910XX (active metabolite of ticagrelor) in the plasma of patients at different time points. Within 24 hr after the administration of 180 mg ticagrelor, the blood concentration of ticagrelor in the HPR group was significantly lower than that in the NPR group, especially during 2, 4, and 6 hr (all p < 0.0001). The plasma concentration of ticagrelor increased rapidly in the NPR group after administration of the loading dose, and reached its peak at 4 hr. The maximal plasma concentration of ticagrelor was 478.1 ± 430.4 ng/mL. After that the plasma concentration of ticagrelor decreased gradually and tended to balance. The plasma concentration of ticagrelor in the HPR group increased gradually within 24 hr and reached its maximum (233.7 ± 195.1 ng/mL) at 24 hr (*Figure 1C*). The plasma concentration of AR-C124910XX at 2, 4, 6, and 8 hr in the HPR group was significantly lower than that in the NPR group (all p < 0.05) (*Figure 1D*). The trend of its change was consistent with that of ticagrelor blood concentration. The effect of ticagrelor on pharmacokinetics mainly occurs in four steps: absorption, distribution, metabolism, and excretion. In our study, we found that the PA rate in the HPR group was higher than that in the NPR group, which was associated with a reduction in plasma concentrations of ticagrelor and AR-C124910XX. This suggests that the pharmacokinetic abnormality of ticagrelor mainly occurs in the absorption stage of the drug.

## Genotyping distribution of *ABCB1*

As a transporter, P-glycoprotein plays an important role in the metabolism and absorption of ticagrelor. *ABCB1* is the gene encoding P-glycoprotein, its SNP may have an important impact on the antiplatelet function of ticagrelor. To detect the difference of *ABCB1* single gene polymorphism between the NPR group and the HPR group, we tested the alleles and gene frequencies of *ABCB1* in the two groups. Functional loci of the *ABCB1* gene polymorphism were rs2032582 (*G2677T/A*), rs1045642 (*C3435T*), and rs1128503 (*C1236T*). Following calculation of the deviation between the observed and theoretical frequencies of *ABCB1* SNP genotypes by the $\chi^2$ test, the results showed that the distribution of *ABCB1* SNP genotypes in the NPR and HPR groups was consistent with the genetic Hardy-Weinberg equilibrium balance (p ≥ 0.05), indicating that the research objects selected in this experiment had good population representativeness (*Table 1*). To prevent class I errors caused by multiple tests, we adjusted the statistical significance level to p < 0.017 after Bonferroni correction. *Table 1* summarized the distributions of alleles and genotype frequencies of the three SNPs in the HPR and NPR groups. There was no significant difference in allele frequency or genotype distribution among the three SNPs of *ABCB1* between the two groups, which suggests that the high reactivity of ticagrelor is not affected by the polymorphic drug transporter ABCB1.

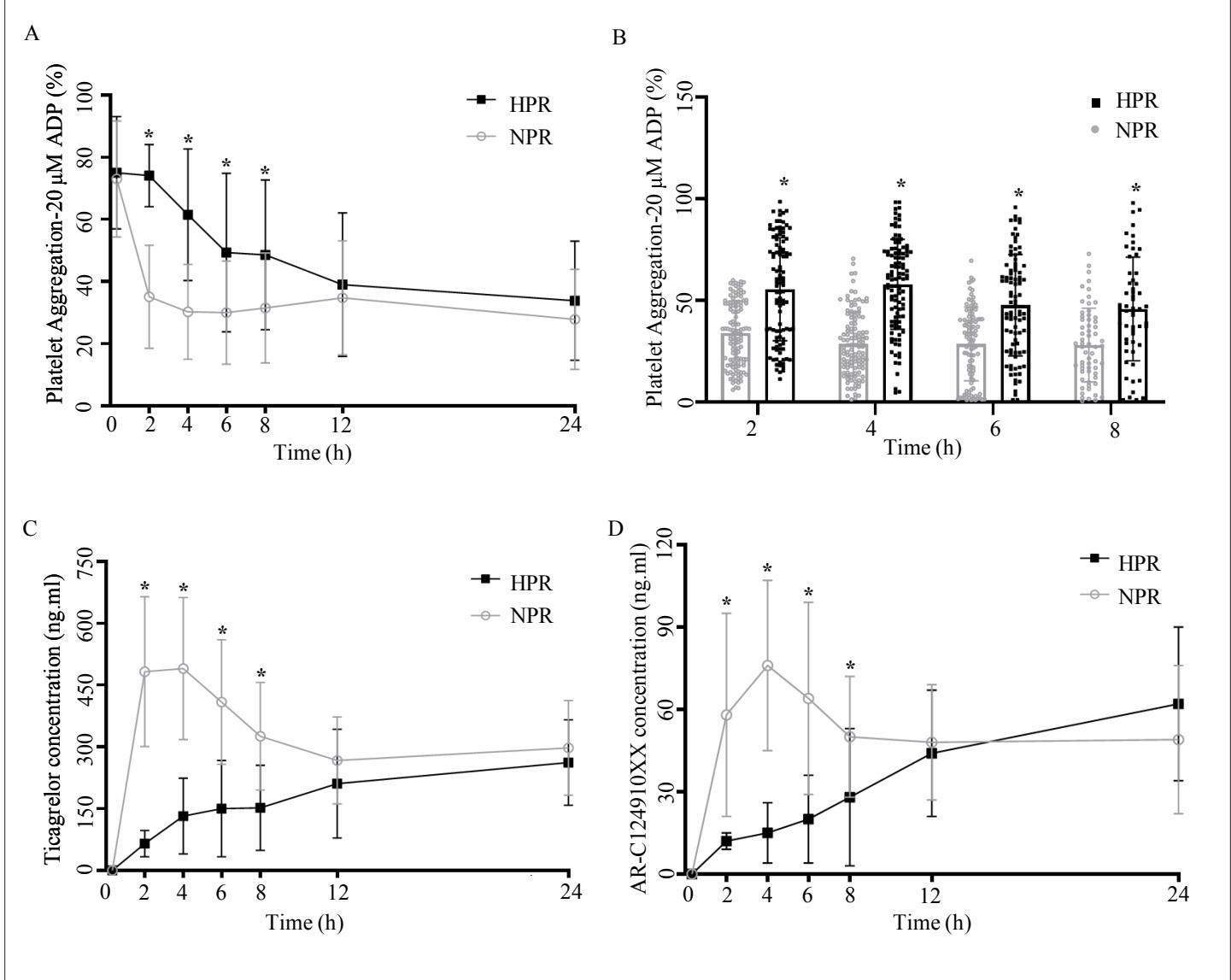

**Figure 1.** Pharmacodynamic and pharmacokinetic assessment of ticagrelor in the high platelet reactivity (HPR) and normal platelet reactivity (NPR) groups. (**A**) Platelet aggregation measured using light transmission aggregometry of line chart at baseline and 2, 4, 6, 8, 12, and 24 hr after the ticagrelor loading dose in patients with NPR and HPR. (**B**) Platelet aggregation of ticagrelor shown in bar graph at 2, 4, 6, and 8 hr after the ticagrelor loading dose in patients with NPR and HPR. (**C, D**) The plasma concentration of ticagrelor (**C**) and its major active metabolite AR-C124910XX (**D**) during the 24 hr following administration of the loading dose of ticagrelor. Values are expressed as the mean. Error bars indicate standard deviation. *p < 0.05 versus NPR.

The online version of this article includes the following source data and figure supplement(s) for figure 1:

**Source data 1.** Twelve-month follow-up results of enrolled normal platelet reactivity (NPR) and high platelet reactivity (HPR) patients.

**Source data 2.** Datasheet of plasma concentration of ticagrelor and its major active metabolite AR-C124910XX at baseline and 2, 4, 6, 8, 12, and 24 hr following administration of the loading dose of ticagrelor in normal platelet reactivity (NPR) and high platelet reactivity (HPR) patients (related to *Figure 1A and B*).

**Source data 3.** Datasheet of platelet aggregation measured using light transmission aggregometry at baseline and 2, 4, 6, 8, 12, and 24 hr after the ticagrelor loading dose in normal platelet reactivity (NPR) and high platelet reactivity (HPR) patients (related to *Figure 1C and D*).

**Source data 4.** General clinical data of enrolled normal platelet reactivity (NPR) and high platelet reactivity (HPR) patients.

**Source data 5.** Comparison of demographic and clinical characteristics between NPR and HPR groups.

**Figure supplement 1.** Flowchart illustrating the recruitment of patients based on the exclusion and inclusion criteria.

**Table 1.** Genotype and allele distributions for the three polymorphisms of *ABCB1* in NPR and HPR groups.

Chi-square test for the genotypic frequency of *ABCB1* gene among NPR and HPR groups. *ABCB1*, ATP binding cassette subfamily B member 1; NPR, normal platelet reactivity; HPR, high platelet reactivity; CI, confidence interval; OR, odds ratio; HWE, Hardy-Weinberg equilibrium. p ≥ 0.05, in accordance with HWE.

| Genotype/ allele | NPR (%) | HWE-P | HPR (%) | HWE-P | OR (95% CI) | p-Value |
|---|---|---|---|---|---|---|
| **Rs1045642** | | | | | | |
| CC | 24 (26.7) | 0.099 | 24 (36.9) | 0.723 | 1 | |
| CT | 53 (58.9) | | 30 (46.2) | | 0.566 (0.285–1.192) | 0.121 |
| TT | 13 (14.4) | | 11 (16.9) | | 0.846 (0.332–2.320) | 0.739 |
| C allele | 101 (56.1) | | 78 (60.0) | | 1 | |
| T allele | 79 (43.9) | | 52 (40.0) | | 0.852 (0.532–1.351) | 0.494 |
| **Rs2032582** | | | | | | |
| GG | 15 (16.7) | 0.360 | 9 (13.8) | 0.111 | 1 | |
| GT | 34 (37.8) | | 29 (44.6) | | 1.422 (0.560–3.567) | 0.473 |
| GA | 11 (12.2) | | 9 (13.8) | | 1.364 (0.435–4.374) | 0.614 |
| TT | 14 (15.6) | | 7 (10.7) | | 0.833 (0.238–2.678) | 0.771 |
| TA | 12 (13.3) | | 8 (12.3) | | 1.111 (0.339–3.578) | 0.865 |
| AA | 4 (4.4) | | 3 (4.6) | | 1.250 (0.265–5.728) | 0.798 |
| G allele | 75 (41.7) | | 56 (43.1) | | 1 | |
| T allele | 74 (41.1) | | 51 (39.2) | | 0.923 (0.568–1.495) | 0.752 |
| A allele | 31 (17.2) | | 23 (17.7) | | 0.994 (0.511–1.874) | 0.985 |
| **Rs1128503** | | | | | | |
| CC | 6 (6.7) | 0.670 | 8 (12.3) | 0.662 | 1 | |
| CT | 51 (56.7) | | 28 (43.1) | | 0.412 (0.123–1.243) | 0.125 |
| TT | 33 (36.7) | | 29 (44.6) | | 0.659 (0.191–1.955) | 0.483 |
| C allele | 63 (35.0) | | 44 (33.8) | | 1 | |
| T allele | 117 (65.0) | | 86 (66.2) | | 1.052 (0.661–1.700) | 0.833 |

The online version of this article includes the following source data for table 1:

**Source code 1.** The original code file of sequencing analysis for tagSNPs of ATP binding cassette subfamily B member 1 (*ABCB1*) gene in normal platelet reactivity (NPR) and high platelet reactivity (HPR) patients (related to *Table 1*).

**Source data 1.** ATP binding cassette subfamily B member 1 (*ABCB1*) tagSNPs from the HapMap database and primer sequences used in genotyping analysis for *ABCB1*.

## HPR group had rich microbiota diversity

The intestinal flora plays an important role in the absorption and metabolism of drugs. To detect whether the intestinal flora has an impact on the absorption and metabolism of ticagrelor, we compared the intestinal flora of patients in the NPR group and the HPR group. Patients treated with antibiotics or probiotics within the last 2 months were excluded. Here, 26 patients from the HPR group and 26 patients from the NPR group were assessed. High-quality 16S rDNA V3-V4 sequences obtained from fecal samples were processed using the Illumina platform. A total of 2,185,831 reads were generated, with a median of 59,271 reads per sample. Following removal of singletons, a total of 975 operational taxonomic units were clustered at 97% sequence similarity.

α-Diversity is the analysis of species diversity in a single sample, including chao1 (*Figure 2A*), observed species diversity index (*Figure 2B*), PD whole tree (*Figure 2C*), and good coverage diversity

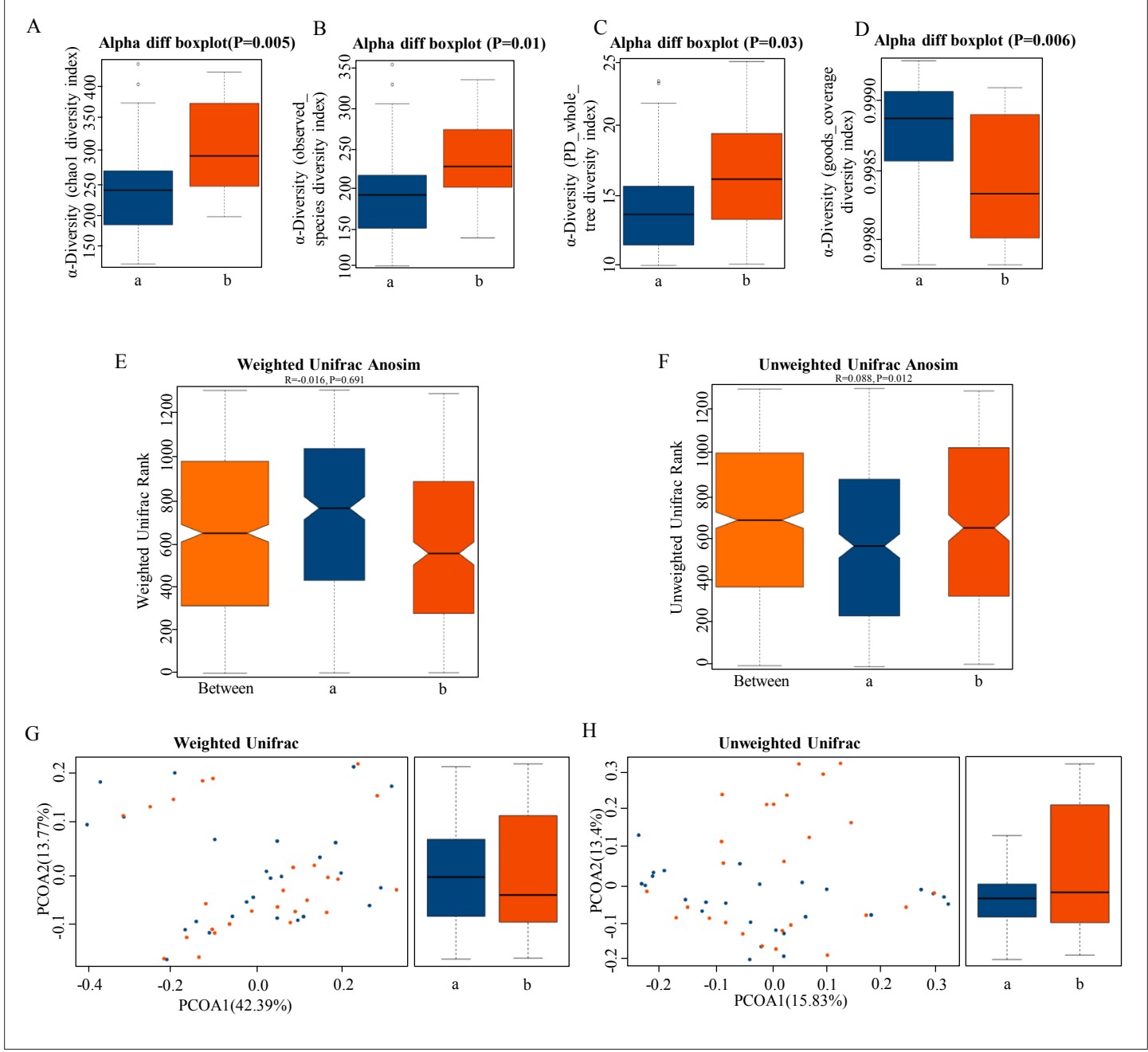

**Figure 2.** The α-diversity and β-diversity indices of the fecal microbiome in the normal platelet reactivity (NPR) and high platelet reactivity (HPR) groups. (**A, B, C, D**) Box plots depict differences in the fecal microbiome diversity indices between the PD and healthy groups according to the chao1 index (**A**), observed species index (**B**), PD whole tree index (**C**), and goods coverage diversity index (**D**) based on the OTU counts. Each box plot represents the median, interquartile range, minimum, and maximum values. (**E, F, G, H**) Unweighted and weighted ANOSIMs and PCOA based on the distance matrix of UniFrac dissimilarity of the fecal microbial communities in the HPR and NPR groups. Box and whiskers distribution of the intra-group unweighted UniFrac distances (**E**) and intra-group weighted UniFrac distances (**F**) calculated for HPR and NPR groups. Respective ANOSIM R values show the community variation between the compared groups, and significant p-values are indicated, as calculated using Tukey post hoc test after Kruskal-Wallis test for multiple comparisons. The axes represent the two dimensions explaining the greatest proportion of variance in the communities. Each symbol represents a sample, and each line connects a pair of samples. a, NPR group (blue); b, HPR group (red). OTU, operational taxonomic unit; ANOSIM, analyses of similarities; PCOA, principal coordinates analysis.

The online version of this article includes the following source data for figure 2:

**Source data 1.** Closing report of 16S rDNA amplicon sequencing related to *Figure 2*.

index (*Figure 2D*). These were significantly higher in the HPR group than in NPR group (p < 0.05), which demonstrated increased species richness in the HPR group. UniFrac uses the evolutionary information of the system to compare species community differences among samples. Significant differences were also found in β-diversity based on the unweighted (qualitative, ANOSIM R = 0.068, p = 0.012, *Figure 2F*) and not the weighted (quantitative, ANOSIM R = –0.016, p = 0.691, *Figure 2E*), indicating that the fecal microbial structure in the HPR group was significantly different from that of the NPR group.

## Dysbiosis of gut microbiota in HPR group patients

Linear discriminant analysis (LDA) effect size (LEfSe) was used to compare the microbiota between the NPR group and the HPR group. *Figure 3A* shows the LDA cluster tree, using the log LDA score cut-off value of 2.0 to identify important taxonomic differences between the two groups. Here, we considered the differences in taxa at the genus level. There were 21 species of bacteria at the genus level in all subjects, and 17 species of bacteria in the HPR group were more abundant than those in the NPR group, including *Bacillus*, *Methylbacterium*, *Staphylococcus*, *Acinetobacter*, and *Brevibacterium*. *Figure 3B* shows an LDA cluster tree. Red indicates bacteria with increased relative abundance in the HPR group, blue indicates bacteria with reduced relative abundance in the NPR group, and the yellow node indicates microbial communities that do not play an important role in both groups. It can be seen from the figure that among the gut microbiota in the HPR group, the most important family/genera are Staphylococcaceae, *Staphylococcus*; Sphingomonadaceae, *Sphingomonas*; Cellulomonadaceae, *Cellulomonas*; and Lactobacillaceae, *Lactobacillus*. These results indicated that there were significant differences in fecal microbiota between the HPR and NPR groups.

## Random forest predictive models

Random forest (RF) is an algorithm based on a classification tree. To evaluate the difference in platelet reactivity between the two groups, we used the abundance groups with significant differences at the genus level obtained by Wilcoxon rank sum test as the input, and established a prediction model based on fecal microflora pedigree by RF. Finally, we obtained the relative importance order of 21 genera to predict the occurrence of HPR. The importance was assessed by the decrease in average accuracy and the Gini index (*Figure 3C*). The genus *Staphylococcus* ranked first among the most important features based on mean decrease accuracy and was found to be 8.0-fold more abundant in the HPR group than the NPR group. *Lactobacillus*, *Methylobacterium*, and *Sphingomonas* were also among the most important genera contributing to classification accuracy. They have a high mean decrease Gini index (>1) and can be considered as candidate biomarkers. The area under the receiver operating characteristic curve was 0.807 (95% confidence interval: 0.612–1.0, 71.4% sensitivity, 88.9% specificity) (*Figure 3D*).

## Predictive function analysis

To clarify the functional characteristics of the gut microbiome in patients with HPR, we used the PICRUST analysis method based on 16S rRNA gene sequence data, and further annotated the gut microbiome metagenomic data in the Kyoto Encyclopedia of Genes and Genomes (KEGG) database. At level 1, the main metabolic pathways with significant differences between the NPR and HPR groups are shown in *Figure 4A*. The relative abundance of the circulatory system and cardiovascular disease was significantly higher in the HPR group than the NPR group. At level 2, there were 24 metabolic pathways involved in the difference between the NPR and HPR groups. The relative abundance of phenylalanine metabolism was the highest, followed by that of cofactor and vitamin metabolism. The relative abundance of the above functions was higher in the NPR group than in the HPR group (*Figure 4B*). In contrast, the HPR group had fewer genes related to apoptosis and ether lipid metabolism than the NPR group.

## Correlation analysis between fecal microbiota composition and metabolic profiles

The metabolites in the $^1$H NMR spectra of blood samples were identified and assigned according to published studies and the human metabolomic database. *Figure 5A* shows the representative $^1$H NMR spectra of the two groups. A total of 32 chemical constituents were identified in this study. We

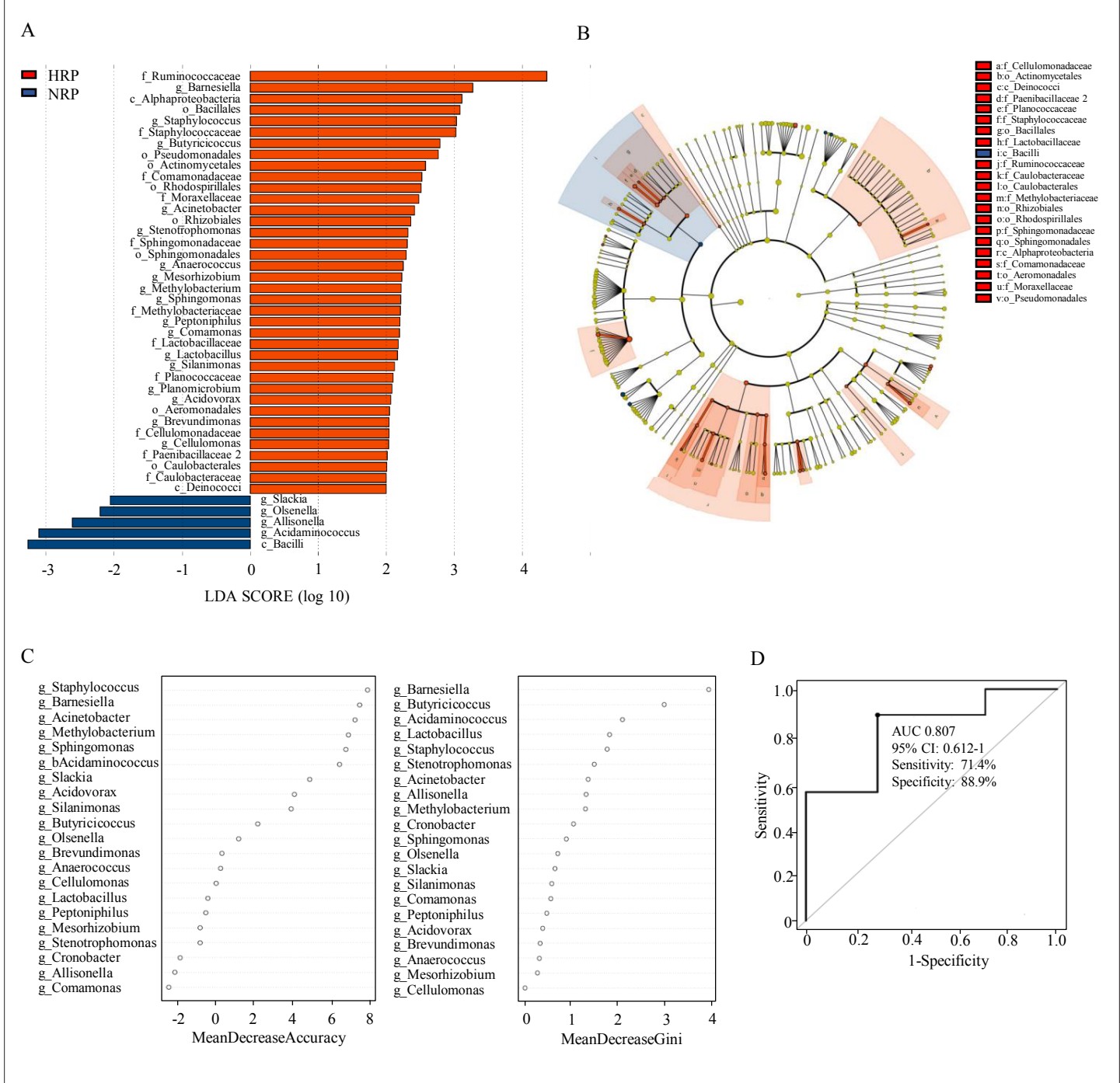

**Figure 3.** Taxonomic differences of fecal microbiota in the normal platelet reactivity (NPR) and high platelet reactivity (HPR) groups. (**A**) Linear discriminant analysis (LDA) effect size (LEfSe) analysis revealed significant bacterial differences in gut microbiota between the NPR (negative score) and HPR (positive score) groups. The LDA scores (log10) >2 and p < 0.05 are listed. (**B**) Cladogram using LEfSe method indicating the phylogenetic distribution of fecal microbiota associated with HPR and NPR groups. (**C**) The predictive model based on genus-level abundance taxa using an RF model. The relative importance of each genus in the predictive model was performed using the mean decreasing accuracy and the Gini coefficient for fecal microbiota. (**D**) ROC curve generated by RF in gut microbiota. The plots shown in the ROC represent the corresponding optimal threshold. RF, random forest; ROC, receiver operating characteristic; AUC, area under the ROC curve; CI, confidence interval.

The online version of this article includes the following source data for figure 3:

**Source data 1.** Closing report of 16S rDNA amplicon sequencing related to *Figure 3*.

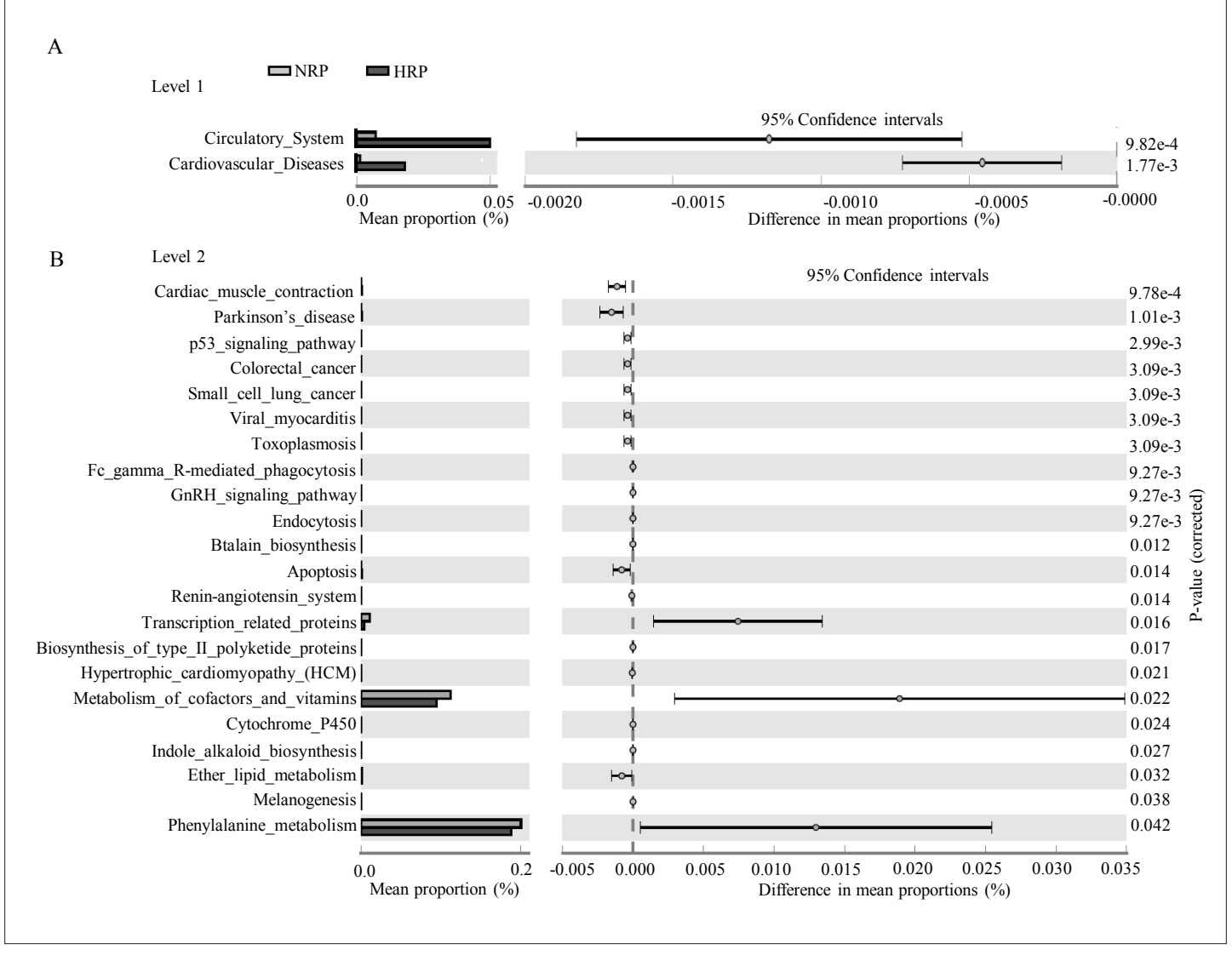

**Figure 4.** Functional predictions for the fecal microbiome of the normal platelet reactivity (NPR) and high platelet reactivity (HPR) groups. The important KEGG pathway of gut microbiota in the HPR and NPR groups was identified using stamp software. White's nonparametric t-test was used to compare the abundance differences between the two groups. The confidence interval was estimated using the percentile bootstrap method (10,000 repetitions). KEGG, Kyoto Encyclopedia of Genes and Genomes; Ko, KEGG homologues; PICRUS, community phylogeny survey by reconstructing unobserved states.

The online version of this article includes the following source data for figure 4:

**Source data 1.** Relative abundance data for prediction of gut microbiota function at level 2 of Kyoto Encyclopedia of Genes and Genomes (KEGG) pathway.

**Source data 2.** Relative abundance data for prediction of gut microbiota function at level 3 of Kyoto Encyclopedia of Genes and Genomes (KEGG) pathway.

used the observed resonance signal phase to determine the relative concentration changes of metabolites in different groups and the color projection of the spectrum to determine the correlation of NMR data between different groups (red corresponds to high correlation [r > 0.6], blue represents no correlation [r < 0.2]). Among them, the content of citrate increased in the HPR group, and the content of valine, glutamate, and salicylate increased in the NPR group. These metabolites may be plasma markers of HPR and may be produced by bacteria or their metabolites. Pearson's correlation analysis explored the relationship between the above six significantly changed metabolites and gut microflora at the genus level (*Figure 5B*). In the generated thermogram, red indicates a positive correlation

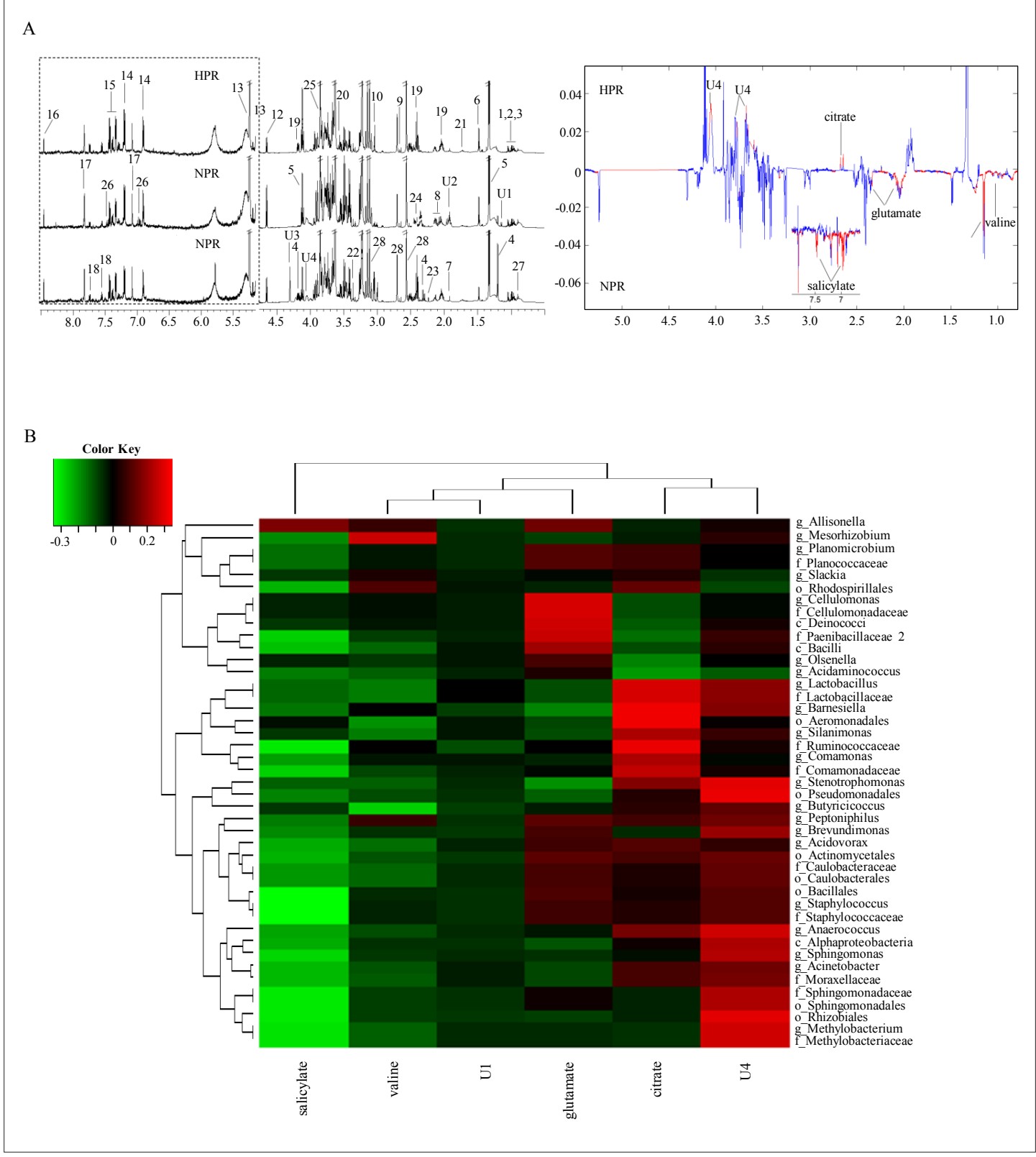

**Figure 5.** [1]H NMR spectra of metabolites from the normal platelet reactivity (NPR) and high platelet reactivity (HPR) groups. (**A**) NMR spectrum and signal assignment diagram. There were 30 samples in the NPR group and 30 samples in the HPR group. The dotted line on the left is the signal spectrum amplified 30-fold. Each number in the figure represents a metabolite. The right is a two color loading graph of the nonparametric test (univariate analysis). 1, leucine; 2, isoleucine; 3, valine; 4, 3-hydroxybutyric acid; 5, lactate; 6, alanine; 7, acetate; 8, glutamate; 9, citrate; 10, creatine;

*Figure 5 continued on next page*

*Figure 5 continued*

11, creatinine; 12, β-glucose; 13, a-glucose; 14, tyrosine; 15, phenylalanine; 16, formate; 17, histidine; 18, tryptophan; 19, pyroglutamate; 20, glycine; 21, lysine; 22, methanol; 23, acetone; 24, succinate; 25, glucitol; 26, salicylate; 27, 2-Hydroxybutyric acid; 28, EDTA; U1, unknown 1; U2, unknown 2; U3, unknown 3; U4, unknown 4. (**B**) Correlation analysis between 16S and significantly changed metabolites. The intensity of the color represents the r value (correlation) (negative score, green; positive score, red).

The online version of this article includes the following source data for figure 5:

**Source code 1.** The original code file of [1]H NMR serum metabolite analysis in normal platelet reactivity (NPR) and high platelet reactivity (HPR) patients.

**Source data 1.** Linear discriminant analysis effect size (LEfSe) analysis was used to analyze the correlation between gut microbiota and metabolites in normal platelet reactivity (NPR) and high platelet reactivity (HPR) patients.

between bacterial taxa and metabolites, while green indicates a negative correlation. Correlation coefficients with statistical differences are listed in *Table 2*, for example, *Lactobacillus*, *Barnesiella*, and *Cellulomonas* positively correlated with citrate and glutamate levels, while *Methylobacterium*, *Sphingomonas*, and *Staphylococcus* negatively correlated with salicylate.

## Platelet reactivity can be transferred by fecal transplantation

Previous studies have shown that antibiotics and probiotics are important factors affecting platelet reactivity in both animal models and clinical trials (*Betancur et al., 2020*; *Budzyński et al., 2016*; *Capkin and Altinok, 2009*; *Kang et al., 2019*). We speculate that the change in platelet reactivity may be related to the change in gut flora with the use of pro/antibiotics. Fecal transplantation is a direct

**Table 2.** Significant correlations between the differential fecal metabolites and microbes in the class, order, family, and genus levels.

Statistical method: Pearson's correlation coefficient; listed correlation coefficients are those with p-value < 0.05.

|  | Glutamate | Citrate | Salicylate |
|---|---|---|---|
| **Class** | | | |
| c_Deinococci | 0.2759636 | – | – |
| **Order** | | | |
| o_Aeromonadales | – | 0.3212488 | – |
| o_Bacillales | – | – | –0.343397 |
| o_Rhizobiales | – | – | –0.307195 |
| o_Sphingomonadales | – | – | –0.315105 |
| **Family** | | | |
| f_Cellulomonadaceae | 0.2813448 | – | – |
| f_Lactobacillaceae | –0.099515 | 0.2799663 | – |
| f_Methylobacteriaceae | – | – | –0.305264 |
| f_Ruminococcaceae | – | 0.3119055 | –0.314255 |
| f_Sphingomonadaceae | – | – | –0.315105 |
| f_Staphylococcaceae | – | – | –0.336272 |
| **Genus** | | | |
| g_Barnesiella | – | 0.3164411 | – |
| g_Cellulomonas | 0.2813448 | – | – |
| g_Lactobacillus | – | 0.2799663 | – |
| g_Methylobacterium | – | – | –0.305264 |
| g_Sphingomonas | – | – | –0.281848 |
| g_Staphylococcus | – | – | –0.336272 |

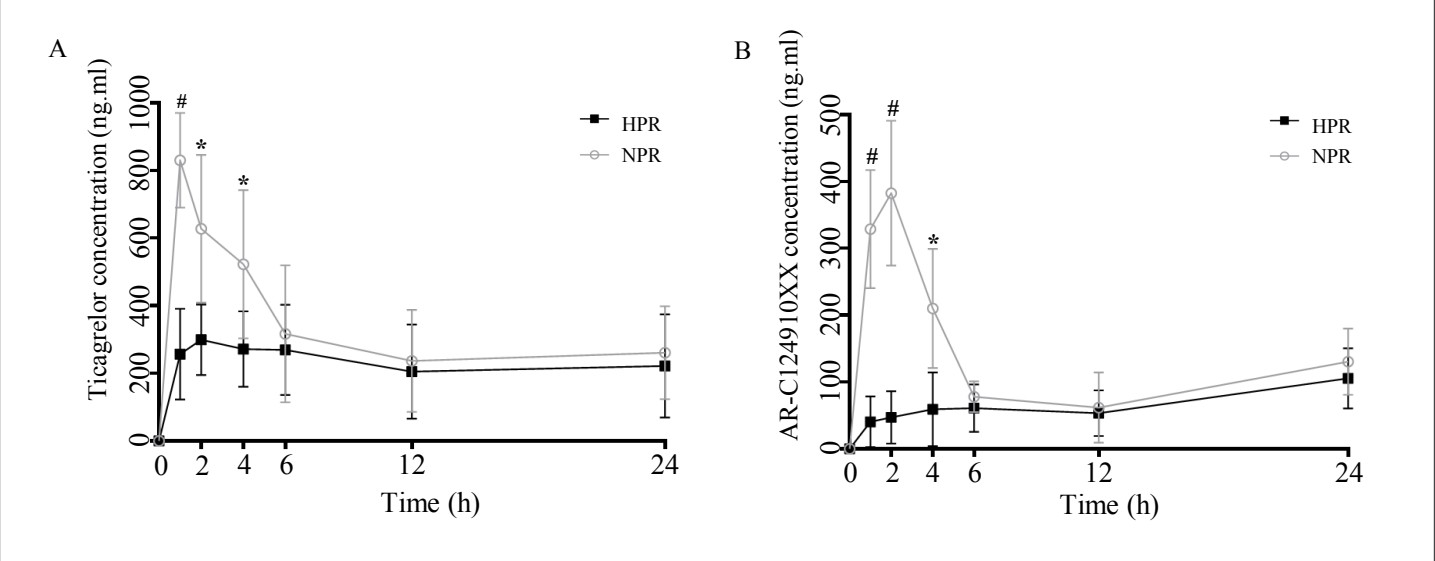

**Figure 6.** Post-transplanted pharmacokinetic assessment of ticagrelor in recipient mice of normal platelet reactivity (NPR) and high platelet reactivity (HPR) groups. The plasma concentration of ticagrelor (**A**) and its major active metabolite AR-C124910XX (**B**) during the 24 hr following administration of ticagrelor loading dose in fecal microbiota transplantation mice. n = 6; values are expressed as the mean. Error bars indicate standard deviation. *p < 0.05 versus NPR, #p < 0.01 versus NPR.

The online version of this article includes the following source data for figure 6:

**Source data 1.** Datasheet of plasma concentration of ticagrelor and its major active metabolite AR-C124910XX in fecal microbiota transplantation mice at baseline and 1, 2, 4, 6, 12, and 24 hr after the ticagrelor loading dose.

experimental method to verify the influence of gut flora. Therefore, in this study, fecal bacteria from HPR and NPR patients were transplanted into microbiota-depleted mice. Plasma concentrations of ticagrelor and AR-C124910XX were measured at different time points after ticagrelor administration in the HPR and NPR groups 8 weeks after transplantation. As expected, the plasma concentration of ticagrelor in the HPR group was lower than that in the NPR group at all time points within 24 hr after ticagrelor administration, and there were significant differences between the two groups at 1, 2, and 6 hr. The plasma concentration of ticagrelor in HPR increased gradually within 24 hr, and the overall concentration was lower than that in the NPR group. The overall change trend of AR-C124910XX in 24 hr was the same as that of the prototype drug (*Figure 6A and B*). Therefore, the change in platelet reactivity may be the mechanism by which antibiotics and probiotics regulate platelet performance.

## Discussion

The present study revealed an interesting finding that gut microbes modulated platelet reactivity after ticagrelor treatment. The genetic determinants of the pharmacokinetics of ticagrelor and gut microbiota have not been previously reported. Therefore, we performed genetic polymorphism analysis and found that there was no difference in ABCB1 gene expression between the NPR and HPR groups. Subsequently, we applied a strategy based on metagenomic and metabolomic analyses to identify the microbial profiles of HPR populations in clinical trials, metabolite signature profiles, and microbial biomarkers for early detection of abnormal platelet responses. Finally, we used fecal transplantation to verify that the manipulation of gut microbiota was sufficient to induce HPR after ticagrelor administration.

Platelet hyperactivity plays an important role in the occurrence and development of cardiovascular diseases. In the past decade, researchers have realized that gut microbiota are involved in the formation of many cardiac metabolic phenotypes and promote the development of atherosclerosis and other diseases (*Duttaroy, 2021*). It has been found that platelets of sterile mice show reduced ADP-dependent PA, indicating that the presence of symbionts sensitizes ADP-induced platelet activation (*Kiouptsi et al., 2020*). The underlying mechanism of these findings may be related to the activation of the innate immune pathway triggered by TLR2 (*Jäckel et al., 2017*) or activation of integrin αIIbβ3

(*Kiouptsi et al., 2020*). Furthermore, the level of functional choline utilization C (cut C) in the gut microbiota can be transmitted through fecal transplantation, thereby enhancing platelet reactivity and thrombosis potential in the recipient (*Skye et al., 2018*). TMAO, an intestinal bacterial by-product, can regulate the high reactivity and clot formation rate of platelets in vivo and increase the risk of thrombosis (*Zhu et al., 2016*).

Presently, the standard of treatment in STEMI patients includes dual antiplatelet therapy consisting of aspirin plus a P2Y12 receptor antagonist. The currently used P2Y12 receptor inhibitors are clopidogrel, prasugrel, and ticagrelor. Moreover, clopidogrel and prasugrel require metabolic activation and irreversibly bind to the P2Y12 receptor, causing prolonged recovery of platelet function. In addition, the efficacy of clopidogrel is reduced in patients with certain genotypes. Ticagrelor is a new oral antiplatelet agent of the cyclopentyltriazolopyrimidine class which acts through the P2Y12 receptor. In contrast to clopidogrel and prasugrel, ticagrelor does not require metabolic activation and binds rapidly and reversibly to the P2Y12 receptor.

Intestinal microbiota is one of the determinants of pharmacokinetics that has been underestimated. When oral drugs enter the digestive tract, they are first exposed to a large number of bacteria and active enzymes secreted by these bacteria, including oxidoreductase and transferase, among others. Therefore, the intestinal biotransformation of drugs may occur before the first-pass effect, affecting the bioavailability and efficacy of drugs. In recent years, the human gut has been found to be closely associated with cardiovascular diseases. Animal experiments have shown that the decrease in left ventricular function and intestinal blood supply caused by myocardial infarction leads to the inhibition of intestinal epithelial tight junction protein and the injury of intestinal mucosa, thereby increasing intestinal permeability. It has been reported that, compared with healthy individuals, the gut microbiota of STEMI patients has higher richness and diversity (*Zhou et al., 2018*). In this study, we analyzed the microbial diversity in fecal samples from the NPR and HPR groups and found that the number of gut microbiota species in the HPR group was significantly higher than that in the NPR group. From species composition analysis, we determined that the concentrations of *Deinococcus* and *Proteus* in the HPR group were significantly higher than those in the NPR group, suggesting that the increase in intestinal harmful bacteria may affect the PA rate. In terms of species abundance, the number of Proteobacteria increased significantly in the NPR group compared with that in the HPR group. Proteobacteria adhere to a large area of gut lining, secreting a large number of enzymatic complexes, such as polysaccharides and fibrin. Moreover, fibrin decomposes fibrinogen in the blood clot and reduces the concentration of plasmin inhibitor in plasma, thus inhibiting platelet activity. This may be another reason for the difference in PA between the HPR and NPR groups.

Next, this study further revealed the relationship between specific metabolite changes and gut bacterial changes. For example, we found that the content of salicylate in the serum of NPR patients increased significantly. In addition, salicylate was negatively correlated with the abundance of *Methylbacillus*, *Sphingomonas*, and *Staphylococcus*. It has long been reported that salicylate inhibits the growth of many bacteria, the production of bacterial virulence factors, and affects the sensitivity of bacteria to drugs (*Sox and Olson, 1989*). Kazama et al. found that salicylate can induce immune thrombocytopenia and inhibit platelet production in megakaryocytes (*Kazama et al., 2015*).

The safety and effectiveness of drugs are key problems in clinical treatment. Studies have shown that intestinal flora plays an important role in pharmacodynamics and pharmacokinetics of drugs (*Kashyap et al., 2017*). Therefore, human intestinal flora may provide a reasonable explanation for individual differences in specific drug reactions (*Vázquez-Baeza et al., 2018*). For example, intestinal microorganisms can affect the metabolism and efficacy of digoxin (*Kumar et al., 2018*) and alter the absorption of vitamin K2, which affects the absorption of warfarin (*Clark et al., 2014*). Our results suggest that 16S analysis of STEMI patients may help identify patients who benefit more from ticagrelor than from clopidogrel treatment. Accurate consideration of the use of pharmaceutical microbiology to reduce drug side effects and improve efficacy by modifying the microbiota would be beneficial in clinical practice. This also reflects the concept of precision medicine, a new method of disease prevention and treatment that takes into account individual genetic differences, living environment, and habits. Although it is challenging to alter human genes, it is relatively easy to modify the human microbiota. The combination of intestinal microbiology, epidemiology, rapid analysis of metabolites and molecular signals, and genetic engineering are expected to aid in the development of a new treatment strategy for STEMI. In the future, it will be necessary to evaluate the specific functions

and potential mechanisms of microbiota, such as platelet reactivity, and to consider microbiota as a potential therapeutic target for cardiovascular diseases.

This study had certain limitations. First, the number of fecal samples selected in this study was small but statistically significant, which can be further confirmed by expanding clinical sample size. Second, the population used in this study represents a certain region and cannot represent the entire Chinese Han population. Third, the 16S method adopted in this study can only detect microorganisms at the genus level. Metagenome analysis was required to identify specific species. Thus, our results need to be verified with a larger sample size and in different populations. Fourth, owing to the small sample size, the incidence of clinically major adverse cardiac events did not increase in the ticagrelor high-response group.

Finally, by establishing a human gut microflora into microbiota-depleted mouse model, the direct effect of gut microbiota components on the regulation of ticagrelor plasma concentration was evaluated to explore the exact mechanism of gut microbiota in platelet response regulation. Our work provides the first direct evidence highlighting the key role of gut microbiota as an important pathogenic factor in high platelet response. Therefore, regulation of gut microbiota should be considered in antiplatelet therapy. Our findings point to a new strategy aimed at preventing the development of high platelet response and reducing cardiovascular risk by restoring the dynamic balance of genetically modified organisms, diet and lifestyle improvements, and early intervention with drugs or probiotics.

## Acknowledgements

Sincere thanks are due to Ms Tong for assistance with the experiments and to Dr Han for valuable discussion.

## Additional information

### Funding

| Funder | Grant reference number | Author |
| --- | --- | --- |
| National Natural Science Foundation of China | 82170297 | Yaling Han |
| National Natural Science Foundation of China | 82070300 | Chenghui Yan |

The funders had no role in study design, data collection and interpretation, or the decision to submit the work for publication.

### Author contributions

Xi Zhang, Formal analysis, Investigation, Writing – original draft, Writing – review and editing; Xiaolin Zhang, Conceptualization, Methodology, Project administration, Supervision, Writing – review and editing; Fangnian Tong, Formal analysis, Writing – original draft; Yi Cai, Methodology, Software; Yujie Zhang, Methodology, Project administration; Haixu Song, Xiaoxiang Tian, Methodology; Chenghui Yan, Investigation, Resources, Supervision, Writing – review and editing; Yaling Han, Conceptualization, Funding acquisition, Project administration, Resources, Supervision, Writing – review and editing

### Author ORCIDs

Yaling Han  http://orcid.org/0000-0003-4569-6737

### Ethics

The use of human sample complies with the Declaration of Helsinki and was approved by the ethics committee of the General Hospital of Shenyang Military Region (2013, No.46: A multicenter, single treatment group, open phase IV study was conducted to evaluate the safety of ticagrelor and describe the incidence of major cardiovascular events in Chinese patients with acute coronary syndrome), and written informed consent was obtained from each subject.

All animal experiments complied with the "Guiding Principles for the Care of Experimental Animals" and "Guidelines for the Care and Use of Experimental Animals" (NIH publication 86-23, revised 1985).

All animals were kept in a pathogen-free environment and fed ad lib. The procedures for care and use of animals were approved by the Committee on the Care and Use of Laboratory Animals of the General Hospital of Northern Theater Command.

## Decision letter and Author response
Decision letter https://doi.org/10.7554/eLife.70240.sa1
Author response https://doi.org/10.7554/eLife.70240.sa2

---

## Additional files

### Supplementary files
• Appendix 1—figure 1—source data 1. Datasheet of plasma concentration of ticagrelor and its major active metabolite AR-C124910XX in single bacterial genera gavaged mice at baseline and 1, 2, 4, 6, 12, and 24 hr after the ticagrelor loading dose (related to *Appendix 1—figure 1A, B*).

• Appendix 1—figure 1—source data 2. Screenshot of original data of platelet aggregation measured using light transmission aggregometry in single bacterial genera gavaged mice at baseline and 1, 2, 4, 6, 12, and 24 hr after the ticagrelor loading dose (Part 1). (related to *Appendix 1—figure 1C*).

• Appendix 1—figure 1—source data 3. Screenshot of original data of platelet aggregation measured using light transmission aggregometry in single bacterial genera gavaged mice at baseline and 1, 2, 4, 6, 12, and 24 hr after the ticagrelor loading dose (Part 2) (related to *Appendix 1—figure 1C*).

• Appendix 1—figure 1—source data 4. Datasheet of platelet aggregation measured using light transmission aggregometry in single bacterial genera gavaged mice at baseline and 1, 2, 4, 6, 12, and 24 hr after the ticagrelor loading dose (related to *Appendix 1—figure 1D*).

• Transparent reporting form

• Source code 1. The original code file of 16S rRNA gene sequencing of gut microbiota c (Part 1).

• Source code 2. The original code file of 16S rRNA gene sequencing of gut microbiota in normal platelet reactivity (NPR) and high platelet reactivity (HPR) patients (Part 2).

• Source code 3. The original code file of 16S rRNA gene sequencing of gut microbiota in normal platelet reactivity (NPR) and high platelet reactivity (HPR) patients (Part 3).

• Source code 4. The original code file of 16S rRNA gene sequencing of gut microbiota in normal platelet reactivity (NPR) and high platelet reactivity (HPR) patients (Part 4).

• Reporting standard 1. STROBE statement.

### Data availability
All data generated or analysed during this study are included in the manuscript and supporting files.

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

## Appendix 1

### Transplantation of single bacterial genera

A single strain of *Sphingomonas* (CICC 10694) or *Staphylococcus* (CICC 10691) was gavaged separately into microbiota-depleted mice (*Methylobacterium* was not used because of the limited experimental time, and slow growth rate). Each group was gavaged with suspensions of living bacteria at a dose of $5 \times 10^8$ CFUs/100 µL in sterile PBS three times a week for 5 weeks. Following oral administration of ticagrelor, we measured the plasma concentrations of ticagrelor and AR-C124910XX, and PA rate in each group. The results showed that compared with the control group (gavaged with sterile PBS), the plasma concentration of ticagrelor and AR-C124910XX in the *Sphingomonas*-treated group decreased significantly at most time points after administration. However, the PA rate in the *Sphingomonas*-treated group was significantly higher than that in the control group at 2 hr after the administration. There was no significant difference in plasma drug concentration and PA between the *Staphylococcus*-treated and control groups (see *Appendix 1— figure 1*). These results suggested that *Sphingomonas* had the most significant impact on the inhibitory effect of ticagrelor in platelet aggregometry.

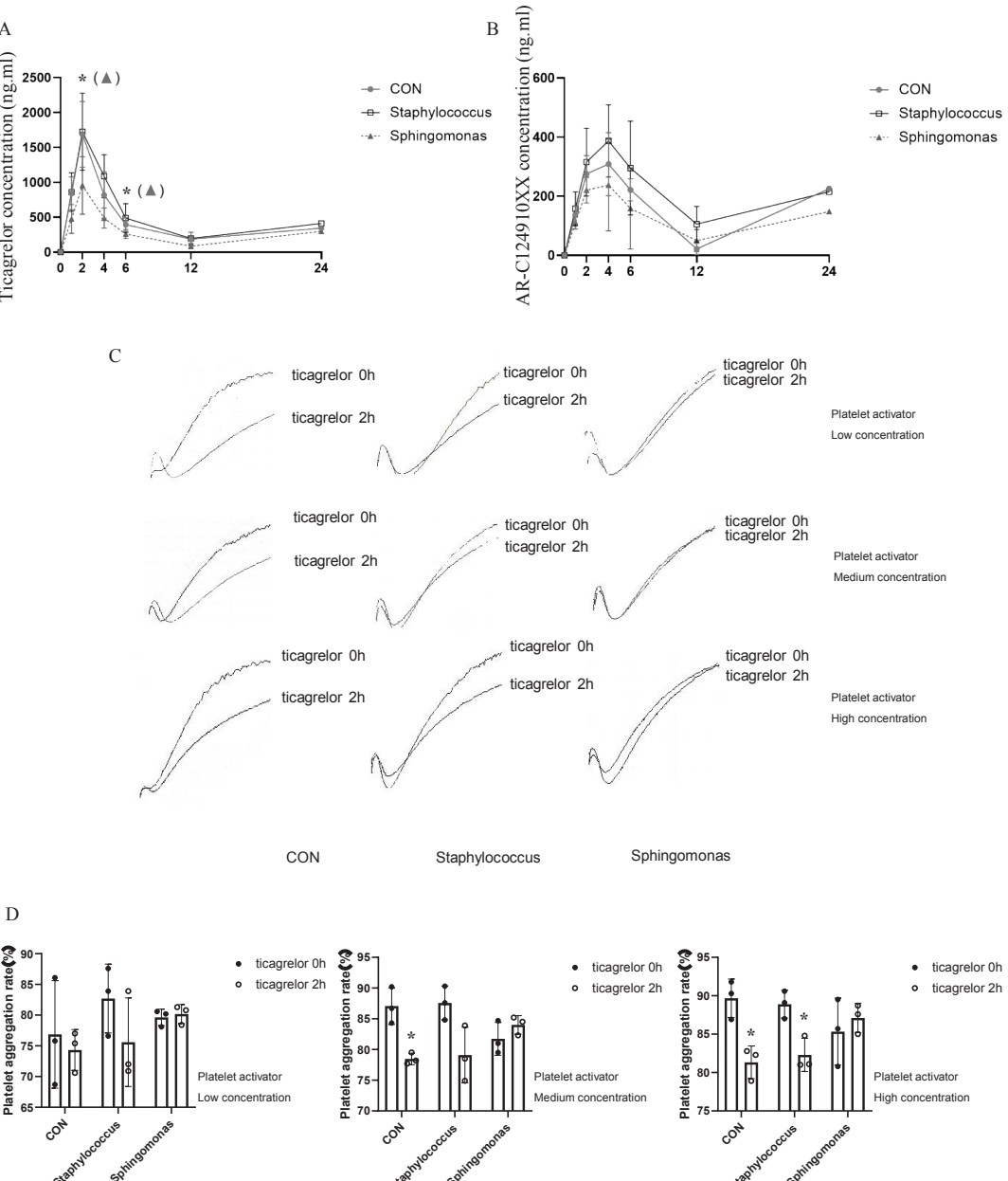

**Appendix 1—figure 1.** Effects of intragastric administration of *Staphylococcus* and *Sphingomonas* on pharmacokinetic assessment of ticagrelor and platelet aggregation. Suspensions of living *Staphylococcus* and *Sphingomonas* were administrated by oral gavage to C57BL/6 mice three times per week at a dose of 5 × 10⁸ CFUs/0.1 mL PBS for 5 weeks. Sterile PBS was used as a control. (**A, B**) Plasma concentration of ticagrelor (**A**) and its major active metabolite AR-C124910XX (**B**) during the 24 hr following administration of the loading dose of ticagrelor. Values are expressed as mean. Error bars indicate standard deviation. *p < 0.05 versus CON. (**C**) Aggregation traces of washed platelets treated with activator were measured by aggregometry. (**D**) Maximum platelet aggregation rate with different concentrations of activator (n = 3, values are expressed as means. Error bars indicate standard deviation. *p < 0.05 versus 0 hr).

The online version of this article includes the following source data for appendix 1—figure 1:

• **Appendix 1—figure 1—source data 1.** Datasheet of plasma concentration of ticagrelor and its major active metabolite AR-C124910XX in single bacterial genera gavaged mice at baseline and 1, 2, 4, 6, 12, and 24 hr after the ticagrelor loading dose (related to *Appendix 1—figure 1A, B*).

• **Appendix 1—figure 1—source data 2.** Screenshot of original data of platelet aggregation

measured using light transmission aggregometry in single bacterial genera gavaged mice at baseline and 1, 2, 4, 6, 12, and 24 hr after the ticagrelor loading dose (Part 1).

• (related to *Appendix 1—figure 1C*).

• **Appendix 1—figure 1—source data 3.** Screenshot of original data of platelet aggregation measured using light transmission aggregometry in single bacterial genera gavaged mice at baseline and 1, 2, 4, 6, 12, and 24 hr after the ticagrelor loading dose (Part 2) (related to *Appendix 1—figure 1C*).

• **Appendix 1—figure 1—source data 4.** Datasheet of platelet aggregation measured using light transmission aggregometry in single bacterial genera gavaged mice at baseline and 1, 2, 4, 6, 12, and 24 hr after the ticagrelor loading dose (related to *Appendix 1—figure 1D*).

