## [Editor Report]

The authors investigated the possible mechanisms of inadequate platelet inhibition with ticagrelor. The high platelet reactivity was found to be associated with lower plasma concentrations of ticagrelor and its active metabolite as compared to the control group. The idea of a two-step research provided key support for clinical findings with experimental data suggesting that gut microbiota dysbiosis may be an important mechanism for the high platelet reactivity in patients treated with ticagrelor. This scientific approach rises new ideas for further research on potential modifications of the gut microbiota as a new therapeutic option in patients with insufficient platelet response to ticagrelor.

---

## [Decision Letter]

**Decision letter after peer review:**

Thank you for submitting your article "Gut microbiota induces high platelet response in patients with ST-segment elevation myocardial infarction after ticagrelor treatment" for consideration by *eLife*. Your article has been reviewed by 2 peer reviewers, and the evaluation has been overseen by a Reviewing Editor and Matthias Barton as the Senior Editor. The following individual involved in review of your submission has agreed to reveal their identity: Jacek Kubica (Reviewer #1).

The reviewers have discussed their reviews with one another, and this letter is to help you prepare a revised submission.

Essential revisions:

1. In the study group and the controls, the authors excluded differences in single nucleotide polymorphisms responsible for ABCB1 gene expression encoding for the intestinal efflux transporter P-glycoprotein (MDR1), which is also the cause of clopidogrel resistance (Simon T, N Engl J Med., 2009; Su J, PLoS One, 2017). However, ticagrelor unlike clopidogrel is not a thienopyridine and acts on P2Y12 via a different inhibitory mechanism and this may be the reason why ticagrelor treatment is superior in the outcome relative to clopidogrel therapy (thienopyridine) in patients with acute coronary syndromes (Wallentin L, N Engl J Med.; 2009). The authors should make clear why the normalization of the control vs treatment group is important in that case. Please explain the mechanistic aspects of the interaction of ticagrelor with P-glycoprotein in more detail and discuss other potential regulators of ticagrelor availability.

2. Microbiota-depletion by antibiotics certainly does not constitute a germ-free mouse model. Germ-free mice can be generated by aseptic hysterectomy or by embryo transfer. Please rephrase this part of the manuscript by exchanging the term "germ-free" to "microbiota-depleted mice". Also, the method description of this part seems not correct since the fecal microbiota transplantation (FMT) was not into patients but into microbiota-depleted mice. The FMT experiment would be far more conclusive if carried out with germ-free mouse model at isolator conditions. In addition to the fecal transplantation experiment into germ-free mouse models, it would be valuable to perform a monocolonization with one of the bacterial genera (Methylobacterium, Sphingomonas and/or Staphylococcus) to test if one of these predictive modelling based hits impairs the inhibitory effect of ticagrelor in ex vivo platelet aggregometry. The authors should make an attempt to conduct this kind of gnotobiotic experiments. These experiments are required to prove the conclusions made by the authors and to move from correlation-based evidence (modelling) to causality.

3. Based on this study outcomes, could microbiota 16S rDNA analyses be suitable as a personalized medicine approach in the treatment options in STEMI? This is likely more feasible than microbiota modification of patients. Please discuss.

4. In support of the microbiota's modifying role on ADP-induced platelet aggregation, platelets of germ-free mice were shown to have a reduced ADP-dependent platelet aggregation (Kiouptsi K, Int J Mol Sci., 2020), indicating that the presence of commensals sensitizes ADP-induced platelet activation. This functional aspect should be discussed.

5. The discussion could be more concise.

---

## [Author Response]

Essential revisions:1. In the study group and the controls, the authors excluded differences in single nucleotide polymorphisms responsible for ABCB1 gene expression encoding for the intestinal efflux transporter P-glycoprotein (MDR1), which is also the cause of clopidogrel resistance (Simon T, N Engl J Med., 2009; Su J, PLoS One, 2017). However, ticagrelor unlike clopidogrel is not a thienopyridine and acts on P2Y12 via a different inhibitory mechanism and this may be the reason why ticagrelor treatment is superior in the outcome relative to clopidogrel therapy (thienopyridine) in patients with acute coronary syndromes (Wallentin L, N Engl J Med.; 2009). The authors should make clear why the normalization of the control vs treatment group is important in that case. Please explain the mechanistic aspects of the interaction of ticagrelor with P-glycoprotein in more detail and discuss other potential regulators of ticagrelor availability.

We thank the reviewer for this comment. Our study was a prospective before-and-after controlled study. In this study, we used multiple factor correction analysis of patient baseline data to correct the error caused by the concentration differences between the samples. We obtained 90 HPR and 65 NPR patients for an in-depth analysis. Furthermore, we used total area/intensity normalization to measure the platelet aggregation rate (PA) of the two groups at different time points. The advantage of this method is that it can eliminate the influence of dilution factors, overcome the problem of different sample concentrations, to make samples comparable, and improve the repeatability and reliability of the experimental results.^1, 2^

I agree with the reviewer’s suggestion that ticagrelor acts on a new class of P2Y12 receptor inhibitors, which is distinct from clopidogrel and prasugrel with respect to its unique mode of inhibitory mechanism. Therefore, we discussed the mechanism of interaction of ticagrelor with P-glycoprotein (MDR1) and other potential regulators of ticagrelor availability.

“Presently, the standard of treatment in STEMI patients includes dual antiplatelet therapy consisting of aspirin plus a P2Y12 receptor antagonist. The currently used P2Y12 receptor inhibitors are clopidogrel, prasugrel, and ticagrelor. Moreover, clopidogrel and prasugrel require metabolic activation and irreversibly bind to the P2Y12 receptor, causing prolonged recovery of platelet function. In addition, the efficacy of clopidogrel is reduced in patients with certain genotypes. Ticagrelor is a new oral antiplatelet agent of the cyclopentyltriazolopyrimidine class which acts through the P2Y12 receptor. In contrast to clopidogrel and prasugrel, ticagrelor does not require metabolic activation and binds rapidly and reversibly to the P2Y12 receptor. “(Page 2 line 70-78)

“The effect of ticagrelor on pharmacokinetics mainly occurs in four steps: absorption, distribution, metabolism, and excretion. In our study, we found that the platelet aggregation rate (PA) in the HPR group was higher than that in the NPR group, which was associated with a reduction in plasma concentrations of ticagrelor and AR-C124910XX. This suggests that the pharmacokinetic abnormality of ticagrelor mainly occurs in the absorption stage of the drug.” (Page 10 line 285-291)

“P-glycoprotein, also known as ABCB1 protein, is one of the adenosine triphosphate binding cassette genes that mostly exist in the apical membrane of the intestinal mucosa, liver, and kidney^3, 4^. It encodes transporters and channel proteins, acts as an efflux pump, and is responsible for intracellular homeostasis. Marsousi N et al^5^., revealed a significant inhibitory effect of the P-glycoprotein inhibitor valspodar on the efflux of ticagrelor, suggesting that P-glycoprotein is involved in the oral disposal of ticagrelor.” (Page 3 line 96-102)

“A large genetic substudy of the PLATO trial showed that CYP2C19 and ABCB1 polymorphisms were independent of the lower rates of cardiovascular death, MI, or stroke observed in patients treated with ticagrelor^6-8^. In addition, a genome-wide association study (GWAS) of patients from the PLATO study identified an SNP associated with the solute carrier organic anion transporter family member 1B1 (SLCO1B1) gene that leads to decreased organic anion transporting polypeptide 1B1 (OATP1B1) activity. Plasma ticagrelor levels were associated with two independent SNPs in the CYP3A4 region and an SNP in the UDP-glucuronosyltransferase 2B7 (UGT2B7) gene^6^. However, the detailed mechanism remains unclear.” (Page 3 line 104-113)

2. Microbiota-depletion by antibiotics certainly does not constitute a germ-free mouse model. Germ-free mice can be generated by aseptic hysterectomy or by embryo transfer. Please rephrase this part of the manuscript by exchanging the term "germ-free" to "microbiota-depleted mice". Also, the method description of this part seems not correct since the fecal microbiota transplantation (FMT) was not into patients but into microbiota-depleted mice. The FMT experiment would be far more conclusive if carried out with germ-free mouse model at isolator conditions. In addition to the fecal transplantation experiment into germ-free mouse models, it would be valuable to perform a monocolonization with one of the bacterial genera (Methylobacterium, Sphingomonas and/or Staphylococcus) to test if one of these predictive modelling based hits impairs the inhibitory effect of ticagrelor in ex vivo platelet aggregometry. The authors should make an attempt to conduct this kind of gnotobiotic experiments. These experiments are required to prove the conclusions made by the authors and to move from correlation-based evidence (modelling) to causality.

We thank the reviewer for pointing this out. We apologize for our failure to clarify the difference between "germ-free mice" and "microbiota-depleted mice"*.* According to your suggestion, we have replaced the term “germ-free mice” with “microbiota-depleted mice” in the manuscript. We also agree with the reviewer’s comments and thank the reviewer for their kind advice. The experiment was supplemented with intragastric administration of a single strain of *Sphingomonas* (CICC 10694) or *Staphylococcus* (CICC 10691) separately in microbiota-depleted mice (*Methylobacterium* was not used because of the limited experimental time, and slow growth rate). Each group was gavaged with suspensions of living bacteria at a dose of 5 × 10^8^ CFUs/100 µL in sterile PBS three times a week for 5 weeks. Following oral administration of ticagrelor, we measured the plasma concentrations of ticagrelor and AR-C124910XX, and PA in each group. The results showed that compared with the control group (gavaged with sterile PBS), the plasma concentration of ticagrelor and AR-C124910XX in the *Sphingomonas*-treated group decreased significantly at most time points after administration. However, the PA in the *Sphingomonas*-treated group was significantly higher than that in the control group at 2 h after the administration. There was no significant difference in plasma drug concentration and PA between the *Staphylococcus*-treated and control groups. These results suggested that *Sphingomonas* had the most significant impact on the inhibitory effect of ticagrelor in platelet aggregometry.

3. Based on this study outcomes, could microbiota 16S rDNA analyses be suitable as a personalized medicine approach in the treatment options in STEMI? This is likely more feasible than microbiota modification of patients. Please discuss.

We thank the reviewer for this highly appreciated comment. We believe that this is a very good point. Consequently, we have revised the Discussion section as follows: (Page 16 line 485-502)

“The safety and effectiveness of drugs are key problems in clinical treatment. Studies have shown that intestinal flora plays an important role in pharmacodynamics and pharmacokinetics of drugs^9^. Therefore, human intestinal flora may provide a reasonable explanation for individual differences in specific drug reactions^10^. For example, intestinal microorganisms can affect the metabolism and efficacy of digoxin^11^ and alter the absorption of vitamin K2, which affects the absorption of warfarin^12^. Our results suggest that 16S analysis of STEMI patients may help identify patients who benefit more from ticagrelor than from clopidogrel treatment. Accurate consideration of the use of pharmaceutical microbiology to reduce drug side effects and improve efficacy by modifying the microbiota would be beneficial in clinical practice. This also reflects the concept of precision medicine, a new method of disease prevention and treatment that takes into account individual genetic differences, living environment, and habits. Although it is challenging to alter human genes, it is relatively easy to modify the human microbiota. The combination of intestinal microbiology, epidemiology, rapid analysis of metabolites and molecular signals, and genetic engineering are expected to aid in the development of a new treatment strategy for STEMI. In the future, it will be necessary to evaluate the specific functions and potential mechanisms of microbiota, such as platelet reactivity, and to consider microbiota as a potential therapeutic target for cardiovascular diseases.”

4. In support of the microbiota's modifying role on ADP-induced platelet aggregation, platelets of germ-free mice were shown to have a reduced ADP-dependent platelet aggregation (Kiouptsi K, Int J Mol Sci., 2020), indicating that the presence of commensals sensitizes ADP-induced platelet activation. This functional aspect should be discussed.

This comment is highly appreciated. We thank the reviewer for their advice. We have added this point and a new reference in the Discussion section. (Page 15 line 431-443)

“Platelet hyperactivity plays an important role in the occurrence and development of cardiovascular diseases. In the past decade, researchers have realized that gut microbiota are involved in the formation of many cardiac metabolic phenotypes and promote the development of atherosclerosis and other diseases^13^. It has been found that platelets of sterile mice show reduced ADP-dependent platelet aggregation, indicating that the presence of symbionts sensitizes ADP-induced platelet activation^14^. The underlying mechanism of these findings may be related to the activation of the innate immune pathway triggered by TLR2^15^ or activation of integrin αIIbβ3^14^. Furthermore, the level of functional choline utilization C (cut C) in the gut microbiota can be transmitted through fecal transplantation, thereby enhancing platelet reactivity and thrombosis potential in the recipient^16^. TMAO, an intestinal bacterial by-product, can regulate the high reactivity and clot formation rate of platelets in vivo and increase the risk of thrombosis^17^.”

5. The discussion could be more concise.

We thank the reviewer for this comment. As suggested by the reviewer, we refined the Discussion section. (Page 14-17, Line 421-520.)

“The present study revealed an interesting finding that gut microbes modulated platelet reactivity after ticagrelor treatment. The genetic determinants of the pharmacokinetics of ticagrelor and gut microbiota have not been previously reported. Therefore, we performed genetic polymorphism analysis and found that there was no difference in ABCB1 gene expression between the NPR and HPR groups. Subsequently, we applied a strategy based on metagenomic and metabolomic analyses to identify the microbial profiles of HPR populations in clinical trials, metabolite signature profiles, and microbial biomarkers for early detection of abnormal platelet responses. Finally, we used fecal transplantation to verify that the manipulation of gut microbiota was sufficient to induce high platelet reactivity after ticagrelor administration.

“Platelet hyperactivity plays an important role in the occurrence and development of cardiovascular diseases. In the past decade, researchers have realized that gut microbiota are involved in the formation of many cardiac metabolic phenotypes and promote the development of atherosclerosis and other diseases^13^. It has been found that platelets of sterile mice show reduced ADP-dependent platelet aggregation, indicating that the presence of symbionts sensitizes ADP-induced platelet activation^14^. The underlying mechanism of these findings may be related to the activation of the innate immune pathway triggered by TLR2^15^ or activation of integrin αIIbβ3^14^. Furthermore, the level of functional choline utilization C (cut C) in the gut microbiota can be transmitted through fecal transplantation, thereby enhancing platelet reactivity and thrombosis potential in the recipient^16^. TMAO, an intestinal bacterial by-product, can regulate the high reactivity and clot formation rate of platelets in vivo and increase the risk of thrombosis^17^.

Presently, the standard of treatment in STEMI patients includes dual antiplatelet therapy consisting of aspirin plus a P2Y12 receptor antagonist. The currently used P2Y12 receptor inhibitors are clopidogrel, prasugrel, and ticagrelor. Moreover, clopidogrel and prasugrel require metabolic activation and irreversibly bind to the P2Y12 receptor, causing prolonged recovery of platelet function. In addition, the efficacy of clopidogrel is reduced in patients with certain genotypes. Ticagrelor is a new oral antiplatelet agent of the cyclopentyltriazolopyrimidine class which acts through the P2Y12 receptor. In contrast to clopidogrel and prasugrel, ticagrelor does not require metabolic activation and binds rapidly and reversibly to the P2Y12 receptor.

Intestinal microbiota is one of the determinants of pharmacokinetics that has been underestimated. When oral drugs enter the digestive tract, they are first exposed to a large number of bacteria and active enzymes secreted by these bacteria, including oxidoreductase and transferase, among others. Therefore, the intestinal biotransformation of drugs may occur before the first-pass effect, affecting the bioavailability and efficacy of drugs. In recent years, the human gut has been found to be closely associated with cardiovascular diseases. Animal experiments have shown that the decrease in left ventricular function and intestinal blood supply caused by myocardial infarction leads to the inhibition of intestinal epithelial tight junction protein and the injury of intestinal mucosa, thereby increasing intestinal permeability. It has been reported that, compared with healthy individuals, the gut microbiota of STEMI patients has higher richness and diversity^18^ In this study, we analyzed the microbial diversity in fecal samples from the NPR and HPR groups and found that the number of gut microbiota species in the HPR group was significantly higher than that in the NPR group. From species composition analysis, we determined that the concentrations of *Deinococcus* and *Proteus* in the HPR group was significantly higher than those in the NPR group, suggesting that the increase in intestinal harmful bacteria may affect the platelet aggregation rate. In terms of species abundance, the number of *Proteobacteria* increased significantly in the NPR group compared with that in the HPR group. *Proteobacteria* adhere to a large area of gut lining, secreting a large number of enzymatic complexes, such as polysaccharides and fibrin. Moreover, fibrin decomposes fibrinogen in the blood clot and reduces the concentration of plasmin inhibitor in plasma, thus inhibiting platelet activity. This may be another reason for the difference in platelet aggregation between the HPR and NPR groups.

Next, this study further revealed the relationship between specific metabolite changes and gut bacterial changes. For example, we found that the content of salicylate in the serum of NPR patients increased significantly. In addition, salicylate was negatively correlated with the abundance of *Methylobacillum*, *Sphingomonas*, and *Staphylococcus*. It has long been reported that salicylate inhibits the growth of many bacteria, the production of bacterial virulence factors, and affects the sensitivity of bacteria to drugs^19^. Kazama et al., found that salicylate can induce immune thrombocytopenia and inhibit platelet production in megakaryocytes^20^.

The safety and effectiveness of drugs are key problems in clinical treatment. Studies have shown that intestinal flora plays an important role in pharmacodynamics and pharmacokinetics of drugs^9^. Therefore, human intestinal flora may provide a reasonable explanation for individual differences in specific drug reactions^10^. For example, intestinal microorganisms can affect the metabolism and efficacy of digoxin^11^ and alter the absorption of vitamin K2, which affects the absorption of warfarin^12^. Our results suggest that 16S analysis of STEMI patients may help identify patients who benefit more from ticagrelor than from clopidogrel treatment. Accurate consideration of the use of pharmaceutical microbiology to reduce drug side effects and improve efficacy by modifying the microbiota would be beneficial in clinical practice. This also reflects the concept of precision medicine, a new method of disease prevention and treatment that takes into account individual genetic differences, living environment, and habits. Although it is challenging to alter human genes, it is relatively easy to modify the human microbiota. dare expected to aid in the development of a new treatment strategy for STEMI. In the future, it will be necessary to evaluate the specific functions and potential mechanisms of microbiota, such as platelet reactivity, and to consider microbiota as a potential therapeutic target for cardiovascular diseases.

This study had certain limitations. First, the number of fecal samples selected in this study was small but statistically significant, which can be further confirmed by expanding clinical sample size. Second, the population used in this study represents a certain region and cannot represent the entire Chinese Han population. Third, the 16S method adopted in this study can only detect microorganisms at the genus level. Metagenome analysis was required to identify specific species. Thus, our results need to be verified with a larger sample size and in different populations. Fourth, owing to the small sample size, the incidence of clinically major adverse cardiac events did not increase in the ticagrelor high-response group.

Finally, by establishing a human gut microflora in GF mouse model, the direct effect of GM components on the regulation of ticagrelor plasma concentration was evaluated to explore the exact mechanism of gut microbiota in platelet response regulation. Our work provides the first direct evidence highlighting the key role of gut microbiota as an important pathogenic factor in high platelet response. Therefore, regulation of gut microbiota should be considered in antiplatelet therapy. Our findings point to a new strategy aimed at preventing the development of high platelet response and reducing cardiovascular risk by restoring the dynamic balance of genetically modified organisms, diet and lifestyle improvements, and early intervention with drugs or probiotics.”

References

1. Chen J, Zhang P, Lv M, Guo H, Huang Y, Zhang Z, Xu FJAc. Influences of Normalization Method on Biomarker Discovery in Gas Chromatography-Mass Spectrometry-Based Untargeted Metabolomics: What Should Be Considered? 2017;89:5342-5348.

2. Misra BJEjoms. Data normalization strategies in metabolomics: Current challenges, approaches, and tools. 2020;26:165-174.

3. von Richter O, Burk O, Fromm M, Thon K, Eichelbaum M, Kivistö KJCp, therapeutics. Cytochrome P450 3A4 and P-glycoprotein expression in human small intestinal enterocytes and hepatocytes: a comparative analysis in paired tissue specimens. 2004;75:172-183.

4. Glaeser HJHoep. Importance of P-glycoprotein for drug-drug interactions. 2011:285-297.

5. Marsousi N, Doffey-Lazeyras F, Rudaz S, Desmeules J, Daali YJF, pharmacology c. Intestinal permeability and P-glycoprotein-mediated efflux transport of ticagrelor in Caco-2 monolayer cells. 2016;30:577-584.

6. Varenhorst C, Eriksson N, Johansson Å, Barratt B, Hagström E, Åkerblom A, Syvänen A, Becker R, James S, Katus H, Husted S, Steg P, Siegbahn A, Voora D, Teng R, Storey R, Wallentin L, JEhj. Effect of genetic variations on ticagrelor plasma levels and clinical outcomes. 2015;36:1901-1912.

7. Nardin M, Verdoia M, Pergolini P, Rolla R, Barbieri L, Marino P, Bellomo G, Kedhi E, Suryapranata H, Carriero A, De Luca G, JPr. Impact of adenosine A2a receptor polymorphism rs5751876 on platelet reactivity in ticagrelor treated patients. 2018;129:27-33.

8. Wallentin L, James S, Storey R, Armstrong M, Barratt B, Horrow J, Husted S, Katus H, Steg P, Shah S, Becker R, JL. Effect of CYP2C19 and ABCB1 single nucleotide polymorphisms on outcomes of treatment with ticagrelor versus clopidogrel for acute coronary syndromes: a genetic substudy of the PLATO trial. 2010;376:1320-1328.

9. Kashyap P, Chia N, Nelson H, Segal E, Elinav EJMCp. Microbiome at the Frontier of Personalized Medicine. 2017;92:1855-1864.

10. Vázquez-Baeza Y, Callewaert C, Debelius J, Hyde E, Marotz C, Morton J, Swafford A, Vrbanac A, Dorrestein P, Knight RJArop, toxicology. Impacts of the Human Gut Microbiome on Therapeutics. 2018;58:253-270.

11. Kumar K, Jaiswal S, Dhoke G, Srivastava G, Sharma A, Sharma VJJocb. Mechanistic and structural insight into promiscuity based metabolism of cardiac drug digoxin by gut microbial enzyme. 2018;119:5287-5296.

12. Clark N, Delate T, Riggs C, Witt D, Hylek E, Garcia D, Ageno W, Dentali F, Crowther M, JJim. Warfarin interactions with antibiotics in the ambulatory care setting. 2014;174:409-416.

13. Duttaroy AJN. Role of Gut Microbiota and Their Metabolites on Atherosclerosis, Hypertension and Human Blood Platelet Function: A Review. 2021;13.

14. Kiouptsi K, Jäckel S, Wilms E, Pontarollo G, Winterstein J, Karwot C, Groß K, Jurk K, Reinhardt CJIjoms. The Commensal Microbiota Enhances ADP-Triggered Integrin αβ Activation and von Willebrand Factor-Mediated Platelet Deposition to Type I Collagen. 2020;21.

15. Jäckel S, Kiouptsi K, Lillich M, Hendrikx T, Khandagale A, Kollar B, Hörmann N, Reiss C, Subramaniam S, Wilms E, Ebner K, Brühl M, Rausch P, Baines J, Haberichter S, Lämmle B, Binder C, Jurk K, Ruggeri Z, Massberg S, Walter U, Ruf W, Reinhardt CJB. Gut microbiota regulate hepatic von Willebrand factor synthesis and arterial thrombus formation via Toll-like receptor-2. 2017;130:542-553.

16. Skye S, Zhu W, Romano K, Guo C, Wang Z, Jia X, Kirsop J, Haag B, Lang J, DiDonato J, Tang W, Lusis A, Rey F, Fischbach M, Hazen SJCr. Microbial Transplantation With Human Gut Commensals Containing CutC Is Sufficient to Transmit Enhanced Platelet Reactivity and Thrombosis Potential. 2018;123:1164-1176.

17. Zhu W, Gregory J, Org E, Buffa J, Gupta N, Wang Z, Li L, Fu X, Wu Y, Mehrabian M, Sartor R, McIntyre T, Silverstein R, Tang W, DiDonato J, Brown J, Lusis A, Hazen SJC. Gut Microbial Metabolite TMAO Enhances Platelet Hyperreactivity and Thrombosis Risk. 2016;165:111-124.

18. Zhou X, Li J, Guo J, Geng B, Ji W, Zhao Q, Li J, Liu X, Liu J, Guo ZJM. Gut-dependent microbial translocation induces inflammation and cardiovascular events after ST-elevation myocardial infarction. 2018;6:66.

19. Sox T, Olson CJAa, chemotherapy. Binding and killing of bacteria by bismuth subsalicylate. 1989;33:2075-2082.

20. Kazama I, Baba A, Endo Y, Toyama H, Ejima Y, Matsubara M, Tachi MJCp, biochemistry : international journal of experimental cellular physiology b, pharmacology. Salicylate inhibits thrombopoiesis in rat megakaryocytes by changing the membrane micro-architecture. 2015;35:2371-2382.